# A Tripartite Efflux System Affects Flagellum Stability in *Helicobacter pylori*

**DOI:** 10.3390/ijms231911609

**Published:** 2022-10-01

**Authors:** Katherine Gibson, Joshua K. Chu, Shiwei Zhu, Doreen Nguyen, Jan Mrázek, Jun Liu, Timothy R. Hoover

**Affiliations:** 1Department of Microbiology, University of Georgia, Athens, GA 30602, USA; 2Microbial Sciences Institute, Yale University, West Haven, CT 06516, USA; 3Department of Microbial Pathogenesis, Yale School of Medicine, New Haven, CT 06536, USA; 4Institute of Bioinformatics, University of Georgia, Athens, GA 30602, USA

**Keywords:** *Helicobacter pylori*, flagellum, flagellar motor, flagellar sheath

## Abstract

*Helicobacter pylori* uses a cluster of polar, sheathed flagella for swimming motility. A search for homologs of *H. pylori* proteins that were conserved in *Helicobacter* species that possess flagellar sheaths but were underrepresented in *Helicobacter* species with unsheathed flagella identified several candidate proteins. Four of the identified proteins are predicted to form part of a tripartite efflux system that includes two transmembrane domains of an ABC transporter (HP1487 and HP1486), a periplasmic membrane fusion protein (HP1488), and a TolC-like outer membrane efflux protein (HP1489). Deleting *hp1486*/*hp1487* and *hp1489* homologs in *H. pylori* B128 resulted in reductions in motility and the number of flagella per cell. Cryo-electron tomography studies of intact motors of the Δ*hp1489* and Δ*hp1486*/*hp1487* mutants revealed many of the cells contained a potential flagellum disassembly product consisting of decorated L and P rings, which has been reported in other bacteria. Aberrant motors lacking specific components, including a cage-like structure that surrounds the motor, were also observed in the Δ*hp1489* mutant. These findings suggest a role for the *H. pylori* HP1486-HP1489 tripartite efflux system in flagellum stability. Three independent variants of the Δ*hp1486*/*hp1487* mutant with enhanced motility were isolated. All three motile variants had the same frameshift mutation in *fliL*, suggesting a role for FliL in flagellum disassembly.

## 1. Introduction

*Helicobacter pylori*, a member of the phylum Campylobacterota, colonizes the human gastric mucosa where it can cause a variety of diseases, including chronic gastritis, peptic and duodenal ulcers, B cell MALT lymphoma and gastric adenocarcinoma [1]. *H. pylori* possesses a cluster of polar, sheathed flagella that it uses to migrate through the viscous mucus layer covering the stomach epithelium. Flagellar-mediated motility is required for host colonization, as non-motile *H. pylori* mutants are unable to colonize animal models [2,3].

The bacterial flagellum consists of three distinct parts referred to as the basal body, hook, and filament. The basal body includes a rotary motor and the flagellar protein export apparatus that transports axial components of the flagellum (e.g., rod, hook, and filament proteins) across the inner membrane [4,5]. Components of the archetypical flagellar motor of *Escherichia coli* and *Salmonella enterica* include the MS ring, C ring, rod, L ring, P ring, and a stator unit comprised of a complex of the MotA and MotB proteins. The stator unit is the torque generator for the motor and is powered by the proton motive force [6]. The MS and C rings form the rotor and transmit torque through the rod and hook to the filament, which acts as a propeller to push the cell forward [7]. The *H. pylori* flagellar motor accommodates up to 18 stators [8], making it the largest bacterial flagellar motor described to date. In addition to the conserved core features of flagellar motors, the *H. pylori* motor contains several accessories not found in the archetypical *E. coli*/*S. enterica* motor [8,9,10].

Most *Helicobacter* species, including *H. pylori*, possess a membranous sheath surrounding the flagellar filament that is continuous with the outer membrane [8]. Despite the continuity of the sheath and outer membrane, protein compositions of these structures may vary somewhat [11,12,13,14,15]. Proposed roles for the *H. pylori* flagellar sheath include protecting the filament subunits from dissociation by gastric acid and adherence to host cells [16]. In *Vibrio cholerae*, the flagellar sheath has been suggested to hide flagellin from Toll-like receptor 5 and thereby prevent triggering of the host innate immune system [17]. Although *H. pylori* flagellins lack the conserved region that is recognized by Toll-like receptor 5 [18,19], the flagellins of *H. pylori* and its close relative *Campylobacter jejuni* are glycosylated with pseudaminic acids, which modulate the host’s immune response in *C. jejuni* infections [20]. Another proposed function of the flagellar sheath is the generation of outer membrane vesicles (OMVs). Rotation of the sheathed flagellum in *Vibrio* species releases OMVs that contain the potent immune response stimulant lipopolysaccharide (LPS) [21,22]. In the symbiosis between *Vibrio fischeri* and the Hawaiian bobtail squid, *Euprymna scolopes*, LPS associated with the OMVs induces apoptotic cell death in the epithelium of the squid light organ, which is required for normal development of the organ [22]. In addition to communicating with the host innate immune system, OMVs can bind bacteriophage, which appears to afford some protection against phage for bacteria [23,24].

Our understanding of flagellar sheath biosynthesis for any bacterial species is limited [16]. To address this gap in our knowledge, we compared the proteins encoded in the genomes of *Helicobacter* species that have flagellar sheaths (FS^+^) against *Helicobacter* species that have unsheathed flagella (FS^−^) to identify proteins with possible roles in sheath biosynthesis or function. Homologs of nearly 40 proteins were found significantly more frequently in the FS^+^ *Helicobacter* species compared to FS^−^ *Helicobacter* species. Four of these proteins, HP1486, HP1487, HP1488 and HP1489, were found exclusively in FS^+^ *Helicobacter* species and appear to be part of a tripartite efflux system. HP1486 and HP1487 are predicted transmembrane domains of an ATP-binding cassette (ABC) transporter, HP1488 is a periplasmic membrane fusion protein (MFP), and HP1489 is an outer membrane efflux protein (OEP). Tripartite efflux systems transport a variety of substrates directly from the cytoplasm to the external medium, including proteins, oligosaccharides, antibiotics, metals, large cations, and various xenobiotics [25,26]. Deleting *hp1486*/*hp1487* and *hp1489* homologs in *H. pylori* B128 reduced motility and the number of flagella per cell. Imaging the motors of the Δ*hp1489* and Δ*hp1486*/*hp1487* mutants by cryo-electron tomography (cryo-ET) revealed many of the cells contained flagellum substructures consisting of only the L ring, P ring, basal disk, and outer disk, resembling flagellar disassembly products (referred to as PL-subcomplexes) reported for other bacteria [27,28]. In addition to the PL-subcomplex, the Δ*hp1489* mutant contained aberrant motors that lacked specific components, which may be flagellum disassembly intermediates. Variants of the Δ*hp1486*/*hp1487* mutant with enhanced motility were isolated, and each of the variants had the same frameshift mutation in *fliL*. The C-terminal domain of FliL forms a ring that encircles the MotB dimer within the stator unit of the *H. pylori* motor [29]. The frameshift mutation occurred near the beginning of the *fliL* coding region, which was surprising since *fliL* is required for motility in *H. pylori* [29]. We hypothesize the mutation in *fliL* results in the expression of a truncated FliL that suppresses the motility defect in the Δ*hp1487*/*hp1486* mutant. Taken together, our observations suggest roles for the HP1489-HP1486 tripartite efflux system and FliL in the stability and disassembly of the *H. pylori* flagellar motor.

## 2. Results

### 2.1. Identification of H. pylori Proteins That Are Found Preferentially in Helicobacter Species with Flagellar Sheaths

Most *Helicobacter* species identified to date possess a flagellar sheath, but several *Helicobacter* species have unsheathed flagella. To identify proteins with possible roles in sheath biosynthesis or function, we searched for homologs of *H. pylori* proteins that were prevalent in FS^+^ *Helicobacter* species but underrepresented in FS^−^
*Helicobacter* species. Using blastp, predicted proteins encoded in the genomes of 35 FS^+^ *Helicobacter* species and 9 FS^−^
*Helicobacter* species were examined for homologs of the 1458 predicted proteins encoded in the *H. pylori* G27 genome. The *H. pylori* G27 genome was chosen as a reference genome for the analysis since G27 is a model for studies of motility and flagellum biosynthesis in *H. pylori*. *Helicobacter* species and strains analyzed are listed in Appendix A. The blastp comparison was performed with an E-value cutoff of 10^−20^. Under the conditions of the null hypothesis (that any protein is equally likely to have a homolog in the FS^+^ species as in the FS^−^ species), the number of homologs in FS^+^ species is expected to follow the hypergeometric distribution. We therefore used the hypergeometric test (equivalent to one-tail Fisher’s exact test) to obtain a *p*-value for each protein. False discovery rate (FDR) [30] and a more conservative Bonferroni correction were used for multiple testing correction. Results from the blastp analysis for all 1458 *H. pylori* G27 proteins are presented in Appendix A. Forty-two proteins that satisfy the Bonferroni-adjusted cutoff for statistical significance (*p*-value < 3.4 × 10^−5^) in FS^+^ *Helicobacter* species are listed in Table 1. An additional 35 proteins qualified as significantly overrepresented in FS^+^ *Helicobacter* species when the less conservative FDR correction was applied (Appendix A).

### 2.2. HP1486, HP1487 and HP1489 Are Required for Robust Motility of H. pylori

To test the predictions from the comparative genomics approach, we targeted some of the identified genes for deletion analysis. Homologs of four proteins, HP1486, HP1487, HP1488 and HP1489, were present in all the FS^+^ *Helicobacter* species and absent in all the FS^−^ *Helicobacter* species (Appendix A), which made the genes encoding these genes good candidates for further investigation. Based on the functional annotations of the proteins and the linkage of the genes encoding them within an operon (Figure 1), these proteins are predicted to form a tripartite efflux system. Absent in the *hp1489*-*hp1486* locus, however, is a gene encoding the ATPase for the ABC transporter, which presumably is present at a different locus in the genome. The *H. pylori* 26695 and G27 genomes encode 19 and 18 predicted proteins that contain an ATP-binding domain of an ABC transporter (pfam00005), respectively. Most of the *H. pylori* genes encoding ATP-binding domain proteins are associated with genes encoding other ABC transporter genes or encode an ABC transporter transmembrane domain, suggesting these genes do not encode the ATPase that functions with HP1486/HP1487 transmembrane domain proteins. A blastp analysis of the sequences of all the ATP-binding domain proteins from *H. pylori* 26695 versus the genomes of FS^+^ and FS^−^ *Helicobacter* species failed to identify a clear candidate for the ATPase of the HP1486/HP1487 ABC transporter that was supported by the protein functional annotation or yielded orthologous homologs.

To determine if the HP1489-HP1486 efflux system has a role in flagellar biosynthesis or function in *H. pylori*, we deleted hp1489 and *hp1487*-*hp1486* in *H. pylori* B128, and also deleted *hp1489* in *H. pylori* G27, using a sucrose-based counter-selection method [47]. Motilities of all three mutants in soft agar medium were reduced 33 to 50% compared to the wild-type parental strains (Figure 2). Introducing a copy of *hp1489* into the *H. pylori* G27 Δ*hp1489* mutant on a shuttle vector rescued motility (Figure 2B), indicating loss of *hp1489* was responsible for the motility defect in the mutant. Examination of the *H. pylori* B128 ∆*hp1487*-*hp1486* and Δ*hp1489* mutants by transmission electron microscopy (TEM) revealed the strains possessed fewer flagella per cell than wild type (Figure 3). Flagella of the Δ*hp1487*-*hp1486* and Δ*hp1489* mutants were sheathed, indicate sheath biosynthesis was not grossly impaired in the mutants. Taken together, these observations suggest the HP1489-HP1486 efflux system is required for normal flagellum biosynthesis and/or motility.

### 2.3. In-Situ Structures of Flagellar Motors in the Δhp1489 and Δhp1487/hp1486 Mutants Reveal Apparent Flagellum Disassembly Products

The impaired motility and reduced number of flagella in the Δ*hp1489* and Δ*hp1487*/*hp1486* mutants suggested a role for the HP1489-HP1486 efflux system in the assembly or stability of the flagellum. To address this hypothesis, we examined the structure of the flagellar motors of the *H. pylori* B128 Δ*hp1489* and Δ*hp1487*/*hp1486* mutants by cryo-ET. For the analaysis, we selected 77 subtomograms of motors from 48 reconstructions of Δ*hp1489* mutant cells and 137 subtomograms of motors from 70 reconstructions of Δ*hp1487*/*hp1486* mutant cells, which we analyzed by subtomogram averaging and classification. The resulting motor structures are presented in Figure 4.

Many of the motors of the Δ*hp1489* and Δ*hp1487*/*hp1486* mutants were normal in appearance and attached to a flagellar filament (Figure 4A). These flagella were presumably functional and responsible for the motility observed with the mutants. Several of the motors of the mutants, however, appeared to be incomplete, and often lacked an associated filament (Figure 4B–D). In contrast, such incomplete motors were rarely observed for the *H. pylori* B128 wild-type strain. To gain a better understanding of the aberrant motors of the Δ*hp1489* and Δ*hp1487*/*hp1486* mutants, we used a multivariate statistical analysis to resolve distinct motor structures that corresponded to the full motor and three specific class averages for the aberrant motors (Figure 4E–H). The number of subtomograms for each class of motor are indicated in Table 2.

In the class 1 motor or full motor, associated hook and flagellar sheath structures were observed (Figure 4A), and the motor contained the core and accessories features reported in previous cryo-ET reconstructions of the intact *H. pylori* motor (Figure 4E,I) [8]. Similar to the complete motor, the class 2 motor had the associated hook and sheath stuctures (Figure 4B,F,J). In addition, core components of the motor (MS ring, C ring, LP ring, rod and flagellar protein export apparatus), the basal disk, and the outer disk were evident in this class of motor (Figure 4F,J). The cage and other motor components normally found near the inner membrane though were absent in this class of motor (Figure 4F,J).

The class 3 motor lacked the associated hook, filament and flagellar sheath, although the outer membrane above the rod was deformed slightly with a convex bulge (Figure 4C,G,K). Core components of the motor, as well as the basal disk and the outer disk, were visible in the second class of motor, but the cage and motor components normally found near the inner membrane were missing (Figure 4G,K).

The class 4 motor also lacked the associated hook, filament and flagellar sheath (Figure 4D). In contrast to the other classes of motors, the class 4 motor lacked most of the core components of the motor and the only visible structures were the LP ring, the basal disk and the outer disk (Figure 4H,L). As with the class 3 motor, the outer membrane above the rod was deformed slightly with a convex bulge (Figure 4H). The flagellar substructure resembled the outer-membrane-associated relic complexes that result from flagellum disassembly and have been observed in a variety of bacterial species [27,28]. These flagellum disassembly products are referred to as PL-subcomplexes, and so we will refer to this class of motor substructure as the PL-subcomplex.

In bacterial species that lack a flagellar sheath, the PL-subcomplex is associated with a crater-like structure in the outer membrane that is sealed at the bottom by the P ring, L ring proteins or some other undefined molecule [27,28]. The *H. pylori* PL-subcomplex is not associated with a crater-like structure in the outer membrane, but instead the outer membrane over the PL-subcomplex is convex (Figure 4H) and resembles the PL-subcomplexes observed in *Vibrio* species [28], which like *H. pylori* have sheathed flagella.

### 2.4. Isolation of Mutants That Suppress the Motility Defect in the Δhp1487/hp1486 Mutant

To identify mutations that suppress the motility defect associated with loss of the HP1489-HP1486 efflux system, we enriched for variants of the *H. pylori* B128 ∆*hp1487*/*hp1486* mutant that had enhanced motility in motility agar medium. Three independent motile variants were isolated (designated ESM_MV1 through ESM_MV3) that had motilities in soft agar medium that approximated that of wild type *H. pylori* B128.

Whole genome sequencing of the Δ*hp1487*/*hp1486* motile variants and the Δ*hp1487*/*hp1486* parental strain identified several phase variations resulting from the deletions or insertions in homopolymeric runs, as well as single nucleotide polymorphisms (SNPs) present in the motile variants that were not in the parental strain (Table 3). All three Δ*hp1487*/*hp1486* motile variants contained the identical frameshift mutation in *fliL* (Table 3). While some of the alleles appeared in more than one motile variant (e.g., a SNP in *murJ*), the mutation in *fliL* was the only allele that occurred in all three motile variants, and moreover, was the only gene that carried a mutation in all three motile variants (Table 3). The genome sequencing data suggested strongly that the mutation in *fliL* was responsible for the enhance mobility of the Δ*hp1487*/*hp1486* motile variants in soft agar medium.

The mutation in *fliL* in the Δ*hp1487*/*hp1486* motile variants was unexpected since *fliL* is reported to be required for motility in *H. pylori* [29]. The *fliL* mutation was a deletion of an A in a homopolymeric run of 9 A residues near the start of the open reading frame of the gene (Figure 5). The frameshift mutation, which was in codon 16, altered the amino acid sequence at the point of the mutation and resulted in a stop codon shortly after the deletion. A couple of nucleotides downstream of the introduced stop codon, however, is a potential start codon in the correct reading frame that would allow for the expression of a truncated FliL protein (Figure 5). Moreover, the ATCATG motif upstream of this potential start codon is a reasonable match for a Shine-Dalgarno sequence and is appropriately positioned relative to the start codon. *H. pylori* B128 FliL is 183 amino acids in length, and the truncated protein resulting from the potential start codon indicated in Figure 5B is 154 amino acid residues in length.

## 3. Discussion

### 3.1. The HP1489-HP1486 Tripartite Efflux System Has an Apparent Role in Stabilizing the H. pylori Flagellar Motor

Using a comparative genomics approach, we found homologs of the HP1489-HP1486 efflux system components were present in all 35 FS^+^ *Helicobacter* species but absent in the 9 FS^−^ *Helicobacter* species that we examined. Deletion of *hp1489* or *hp1487/hp1486* in *H. pylori* resulted in reduced motility as assessed by the ability of the mutant strains to migrate from the point of inoculation in soft agar medium (Figure 2). In the case of the *H. pylori* G27 Δ*hp1489* mutant, motility was restored by introducing a copy of *hp1489* into the mutant on the shuttle vector pHel3 (Figure 2), providing strong evidence that the loss of HP1489 was responsible for the motility defect in the original Δ*hp1489* mutant.

Classification of the motor structures from the Δ*hp1489* and Δ*hp1487/hp1486* mutants revealed many of them were missing the cage and other components associated with the inner membrane (Figure 4F,G). At least three distinct classes of aberrant motors were apparent, one of which resembled the PL-subcomplex observed from flagellum disassembly in various bacterial species [27,28]. The relationship between the *H. pylori* PL-subcomplex and the other two classes of aberrant motors is unclear. The different classes of motors may represent intermediates in the flagellum disassembly pathway. While it is possible the aberrant motors are assembly intermediates, this seems less likely. Qin and co-workers examined the ultrastructure of flagellum assembly intermediates in *H. pylori* using cryo-ET to visualize over 300 flagellar motors. The earliest flagellum assembly intermediate Qin and co-workers observed had progressed through rod assembly, and this intermediate contained the cage and the other motor components present in the mature *H. pylori* flagellum [8]. This argues against the aberrant motors in the Δ*hp1489* mutant being assembly intermediates since they lack the cage.

Two significant questions concerning the HP1489-HP1486 efflux system are: (i) what is the substrate transported by the efflux system; and (ii) how does the efflux system stabilize the flagellum? Regarding the first question, tripartite efflux systems transport a diverse range of proteins, xenobiotics, antibiotics, metal ions and other small molecules [25,26]. While the substrate for the HP1489-HP1486 efflux system is not known, van Amsterdam and co-workers reported that disrupting *hp1489* in *H. pylori* 1061 resulted in increased sensitivity to ethidium bromide [48]. It is not clear from the results of van Amsterdam and co-workers whether the HP1489-HP1486 efflux system transports ethidium bromide, or if loss of the efflux system increases the sensitivity of the *hp1489* mutant to ethidium bromide by altering the permeability of the outer membrane. Regarding the second question, the HP1489-HP1486 efflux system may stabilize the flagellum by transporting a molecule that would otherwise accumulate in the cytoplasm of inner membrane and destabilize the flagellum. Alternatively, the transported molecule may be targeted to the outer membrane where it is needed to stabilize the flagellum. It is possible the structure of the HP1489-HP1486 efflux system itself has a role in stabilizing the flagellum. For example, the efflux system may interact with components of the flagellar motor to stabilize the flagellum. Identifying the ATPase that functions with HP1487 and HP1486 will likely prove helpful for examining the validity of the above models for how the efflux system stabilizes the flagellum. The HP1489-HP1486 homologs in *Helicobacter anseris* MIT 04-9362 are encoded in an operon that contains a gene encoding a predicted ATPase for an ABC transport system. HP0853 is an orphan ATPase (i.e., the gene encoding the protein is not associated with genes encoding transmembrane components of an ABC transport system) and an ortholog of the *H. anseris* ATPase. We are currently investigating whether HP0853 is the ATPase that functions with HP1487 and HP1486.

The occurrence of the identical frameshift mutation in *fliL* in all three Δ*hp1487*/*hp1486* motile variants strongly suggests the *fliL* mutation is responsible for suppressing the motility defect in the parental Δ*hp1487*/*hp1486* mutant (Table 2). This result was unexpected since *fliL* was reported to be required for motility in *H. pylori* [29]. As indicated in Figure 5, the frameshift mutation in *fliL* is predicted to result in the expression of a short polypeptide corresponding to the first 16 amino acids of the FliL N-terminus plus an additional 12 amino acid residues. The frameshift mutation could also result in the expression of an N-terminal truncation of FliL that is missing the first 29 amino acid residues (Figure 5). It is possible that both truncated FliL proteins are expressed and needed to suppress the motility defect of the Δ*hp1487*/*hp1486* mutant. We are currently working to determine if one or both of the truncated FliL proteins are required for suppressing the motility defect of the Δ*hp1487*/*hp1486* mutant. We hypothesize that the mutant *fliL* allele suppresses the motility defect in the Δ*hp1487*/*hp1486* mutant by preventing or limiting flagellum disassembly, and studies are planned to examine the validity of this hypothesis. If our hypothesis tests valid, this would implicate FliL in playing a role in flagellum disassembly in *H. pylori* and perhaps other bacteria. Such a finding would be very exciting as little is known regarding the proteins and factors that control flagellum disassembly.

### 3.2. Identification of Genes Potentially Involved in Sheath Formation

Although many bacteria possess flagellar sheaths, little is known about the proteins and other factors that contribute to sheath biosynthesis in any bacterial species. To address this gap in our knowledge, we searched for *H. pylori* proteins that are conserved in FS^+^ *Helicobacter* species but are absent or underrepresented in FS^−^ *Helicobacter* species with the idea that some of these proteins have roles in sheath biosynthesis or function. A caveat of the approach is the FS^+^ and FS^−^ *Helicobacter* species segregate into distinct phylogenic groups [49], which means some of the proteins identified as specifically occurring in FS^+^ *Helicobacter* species may be shared among the species of that group even though they have roles unrelated to the sheath, flagellum or motility.

Despite these limitations, the comparative genomics approach enabled the identification of candidates for proteins that have roles in sheath biosynthesis. One such candidate is the cardiolipin synthase ClsC [43]. Cardiolipin is a phospholipid that accumulates at the cell pole and septal regions in rod-shaped bacteria due to its ability to form microdomains that exhibit a high intrinsic curvature and thus have a lower energy upon localization to negatively curved regions of the membrane [50,51]. Given that the flagellar sheath is a long narrow tube with a high degree of negative curvature, it is reasonable to postulate that cardiolipin accumulates in the sheath. Consistent with this hypothesis, preparations of sheathed flagella from *H. pylori* G27 contain significant amounts of cardiolipin [52]. Thus, FS^+^ *Helicobacter* species may need ClsC to satisfy an increased demand for cardiolipin in sheath biosynthesis.

Some of the proteins found preferentially in FS^+^ *Helicobacter* species may have roles in mitigating the action of the sheathed flagellum. Rotation of the sheathed flagellum in *Vibrio* species is a major source of OMVs [21,22], and the generation of these OMVs is important in communication between *V. fischeri* and its host and symbiotic partner *E. scolopes* [22]. In contrast, *H. pylori* has presumably evolved to avoid surveillance by the host innate immune system. Several of the proteins found preferentially in FS^+^ *Helicobacter* species enzymes involved in LPS biosynthesis or modification (Table 1), which contribute to the low endotoxin and immunological activity of *H. pylori* LPS [53,54,55]. These LPS modification enzymes may be a strategy that FS^+^ *Helicobacter* species use to avoid activating the host innate immune response by OMVs released from the bacterium during the rotation of the sheathed flagellum.

Several of the proteins conserved in FS^+^ *Helicobacter* species belong to the SLR family (Table 1), members of which contain multiple copies of a degenerate amino acid repeat motif that is characteristic of eukaryotic Sel1 proteins [56]. Although the physiological roles of the *H. pylori* SLR proteins are unclear, some of them bind β-lactam compounds [37,38] and have been suggested to interact with immunomodulatory peptidoglycan fragments to affect the innate immune response [57]. Rotation of the *H. pylori* sheathed flagellum may result in the release of peptidoglycan fragments from the periplasmic space, and FS^+^ *Helicobacter* species may use the peptidoglycan-binding SLR proteins to avoid eliciting a host immune response.

## 4. Materials and Methods

### 4.1. Bacterial Strains and Growth Conditions

*Escherichia coli* DH5α was used for cloning and plasmid construction, and was grown in LB broth or agar medium. Medium was supplemented with ampicillin (100 μg/mL), chloramphenicol (30 μg/mL) or kanamycin (30 μg/mL) when appropriate. *H. pylori* strains were grown at 37 °C on tryptic soy agar (TSA) supplemented with 5% heat-inactivated horse serum (TSA-HS) under an atmosphere consisting of 10% CO_2_, 4% O_2_, and 86% N_2_. For liquid cultures, *H. pylori* strains were grown at 37 °C with shaking in Mueller-Hinton broth (Becton, Dickinson and Company, Sparks, MD, USA) (MHB) supplemented with 5% heat-inactivated horse serum (Gibco, Penrose, Aukland, New Zealand) (MHB-HS) under an atmosphere consisting of 5% CO_2_, 10% H_2_, 10% O_2_ and 75% N_2_. Kanamycin (30 μg/mL) or chloramphenicol (30 μg/mL) was added to the medium used to culture *H. pylori* when appropriate.

### 4.2. Strain Construction

Primers used for PCR are listed in Appendix A and strains used in this study are listed in Table 4. Genomic DNA (gDNA) from *H. pylori* G27 was purified using the Wizard genomic DNA purification kit (Promega, Madison, WI, USA) and used as the template to construct all deletion mutants. Target sequences from gDNA were amplified using Phusion polymerase (New England Biolabs, Ipswich, MA, USA). To facilitate cloning into the pGEM-T Easy vector (Promega) amplicons were incubated with *Taq* polymerase (Promega) at 72 °C for 10 min to add 3′-A overhangs.

*H. pylori* B128 *hpb128_199g42* (*hp1489* homolog) and *hpb128_199g40* along with *hpb128_199g39* (*hp1487* and *hp1486* homologs) were deleted as follows. Regions flanking the target genes were amplified from gDNA using primer pairs P117/P118 plus P119/P120 for *hp1489*, and P177/P178 plus P179/P180 for *hp1487*/*hp1486*. The primer pairs introduced complementary sequence to the ends of the amplicons that were used for overlapping PCR. The complementary sequences also included NheI and XhoI restriction sites for a subsequent cloning step. After joining the flanking regions for each target gene by PCR SOEing, the resulting amplicons were cloned into pGEM-T Easy to generate the plasmids pJC049 (carried flanking sequences for *hp1489*), and pJC076 (carried flanking sequences for *hp1487*/*hp1486*). Plasmid pJC038 carries a kan^R^*-sacB* cassette in which *sacB* was under control of the *H. pylori ureA* promoter [43]. The kan^R^-*sacB* cassette from pJC038 was introduced into unique NheI and XhoI sites in plasmids pJC049 and pJC076 to generate the suicide plasmids pJC051 and pJC080, respectively. The suicide plasmids were introduced by natural transformation into *H. pylori* B128 or *H. pylori* G27 and transformants were screened for kanamycin resistance. The kan^R^-*sacB* cassette in the target genes was removed by transforming the kanamycin-resistant isolates with plasmids pJC049 or pJC076, and transformants in which the kan^R^-*sacB* cassette was replaced with the unmarked deletion were isolated following a sucrose-based counter-selection [47]. Regions flanking the targeted genes were amplified by PCR to identify transformants in which the target gene had been deleted, and the resulting amplicons were sequenced (Eton Biosciences) to confirm the target gene had been deleted.

### 4.3. Motility Assay

Motility was evaluated using a semisolid medium containing MHB-HS, 20 mM MES (2-(4-morpholino)-ethane sulfonic acid) (pH 6.0), 5 μM FeSO_4_ and 0.4% Noble agar. A minimum of three biological samples and three technical replicates were used to assess the motility of each strain. *H. pylori* strains grown on TSA supplemented with 10% heat-inactivated horse serum for 4 days were stab-inoculated into the motility agar and incubated at 37 °C under an atmospheric condition consisting of 10% CO_2_, 4% O_2_ and 86% N_2_. The diameters of the resulting swim halos were measured 7 d post-inoculation. The two-sample *t* test was used to determine statistical significance for the results.

### 4.4. Complementation of Δhp1489 Mutation

Primers 129 and 130 were used to amplify a ~500 bp region immediately upstream of the start codon for *hp1491* from *H. pylori* G27 gDNA, which included the promoter for the gene. Primers 131 and 132 were used to amplify *hp1489* from *H. pylori* G27 gDNA. The primer pairs introduced complementary sequence to the ends of the amplicons that were used for overlapping PCR. The complementary sequences also included BamHI and XhoI restriction sites for a subsequent cloning step. After joining the flanking regions for each target gene by PCR SOEing, the resulting amplicons were cloned into pGEM-T Easy to generate plasmid pJC083. A BamHI-XhoI fragment bearing the *hp1491* promoter fused to *hp1489* (P_hp1491_-*hp1489*) was excised from pJC083 and cloned into the *H. pylori* shuttle vector pHel3 [60]. The resulting plasmid, pHel1489, was introduced into HP122 (*H. pylori* G27 Δ*hp1489*) by natural transformation to generate strain HP128.

### 4.5. Isolation of Variants of ∆hp1487/hp1486 Mutant with Enhanced Motilities

Enrichment of variants of the ∆*hp1487*/*hp1486* mutant with enhanced motility in soft agar medium was accomplished by six rounds of inoculating the mutant in the motility agar medium, allowing the bacteria to migrate from the point of inoculation, picking cells from the edge of the swim halo, and inoculating fresh motility agar medium. Following the last iteration, cells from the edge of the swim halo were streaked onto TSA-HS to obtain single colonies. Three independent enrichments were done and a single motile isolate from each enrichment was saved and characterized.

### 4.6. Genome Sequencing and Analysis

For whole genome resequencing, *H. pylori* gDNA was sonicated to approximately 500-bp fragments in a total of 500 ng. Genomic libraries were prepared using the NEBNext Ultra II DNA library prep kit for Illumina (New England Biolabs), including end-repair, adaptor ligation, and addition of an individual i5/i7 index primer. Illumina sequencing of the genomic libraries was done by Genewiz from Azenta Life Sciences. Sequence analysis was performed using the *breseq* computational pipeline [61]. A reference genome was constructed by mapping reads on an NCBI genome for *H. pylori* B128 (Accession no.: GCA_007844075.1). Reads were mapped to the reference genome, and a minimum variant frequency of 0.8 was used to call variants and identify SNPs. Variants and SNPs in the strains were compared manually to identify ones that were present only in the suppressor strains.

### 4.7. Transmission Electron Microscopy

*H. pylori* strains were grown to late-log phase (OD_600_ ~1.0) in MHB-HS. Cells from 1 mL of culture were pelleted by centrifugation (550× *g*) then resuspended in 125 μL of 0.1 M phosphate-buffered saline. Cells were fixed by adding 50 μL of 16% EM grade formaldehyde and 25 μL of 8% EM grade glutaraldehyde to the cell resuspension. Following incubation at room temperature for 5 min, 10 μL of the cell suspension were applied to a 300 mesh, formvar-coated copper grid and incubated at room temperature for 5 min. The cell suspension was wicked off the grids using a filter paper, and the grids were washed 3 times with 10 μL of water. Cells were stained by applying 10 μL of 1% uranyl acetate to the grids for 30 s. After removing the stain with filter paper, the grids were washed three times with 10 μL of water and then air-dried. Cells were visualized using a JEOL JEM 1011 transmission electron microscope. The number of flagella per cell (n = 50) were determined for each strain. A Mann-Whitney U test was used to identify statistically significant differences in number of flagella per cell for the various *H. pylori* strains.

### 4.8. Sample Preparation for Cryo-EM Observation

*H. pylori* strains were cultured on plates for 4 d. The cells on the agar surface were collected then washed with PBS buffer. Colloidal gold solution (10 nm diameter) was added to the diluted *H. pylori* samples and then deposited on a freshly glow-discharged EM grid for 30 s. The grid was blotted with filter paper and rapidly plunge-frozen in liquid ethane in a homemade plunger apparatus as described [62].

### 4.9. Cryo-ET Data Collection and Image Processing

The frozen-hydrated specimens of *H. pylori* strains were transferred to Titan Krios electron microscope (Thermo Fisher) equipped with a 300 kV field emission gun and a Direct Electron Detector. The data was acquired automatically with Tomography from Thermo Fisher. A total dose of 50 e^−^/Å^2^ is distributed among 35 tilt images covering angles from −51° to +51° at tilt steps of 3°. The tilt series were aligned using gold fiducial markers and volumes reconstructed by the weighted back-projection method using IMOD and Tomo3d software to generate tomograms [63,64]. In total, 207 tomograms from *H. pylori* B128 WT cells, 48 tomograms from *H. pylori* B128 Δ*hp1489* cells and 70 tomograms from the *H. pylori* B128 Δ*hp1487*/*hp1486* double mutant cells were generated.

### 4.10. Sub-Tomogram Analysis with i3 Packages

Bacterial flagellar motors were detected manually using the i3 program [65,66]. The orientation and geographic coordinates of selected particles were then estimated. In total, 511, sub-tomograms of *H. pylori* B128 WT, 77 sub-tomograms of *H. pylori* B128 Δ*hp1489* mutant and 137 sub-tomograms of the *H. pylori* B128 Δ*hp1487*/*hp1486* double mutant flagellar motors were used to sub-tomogram averaging analysis. The i3 tomographic package was used to generate class averages of the motor as described [62].

## Figures and Tables

**Figure 1 ijms-23-11609-f001:**
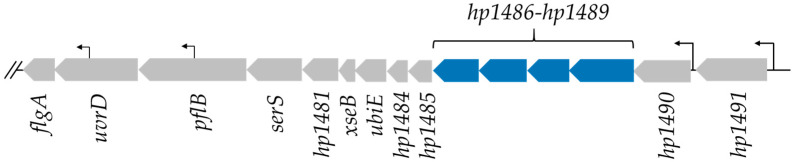
*H. pylori* 26695 operon that contains genes encoding the HP1486-HP1489 tripartite efflux system. Only the first 15 genes of the potential operon are shown. The larger arrows indicated the predicted open reading frames and are drawn approximately to scale. *hp1489*, *hp1488*, *hp1487*, and *hp1486* are indicated in blue. The operon includes two flagellar genes, *flgA* (encodes chaperone for the P ring protein) and *pflB* (paralyzed flagella protein B). Other genes of known function are *uvrD* (encodes DNA helicase II), *serS* (encodes seryl-tRNA synthetase), xseB (encodes exodeoxyribonuclease VII small subunit), and *ubiE* (encodes demethylmenaquinone methyltransferase). The remaining genes encode proteins of unknown function, although *hp1491* is predicted to encode a phosphate transporter. The smaller arrows indicate identified transcriptional start sites that were identified from the *H. pylori* 26695 transcriptome [46].

**Figure 2 ijms-23-11609-f002:**
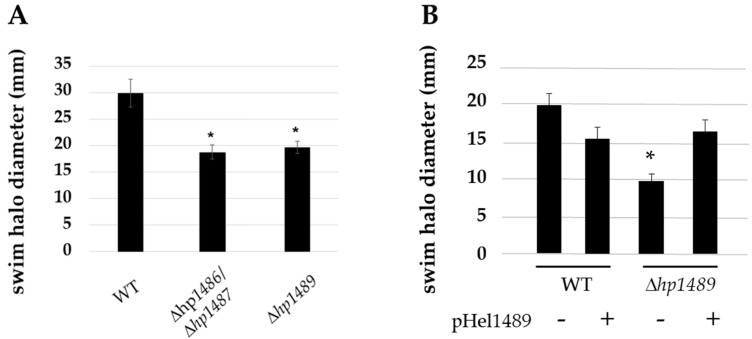
Motility of *H. pylori* Δ*hp1486*/*hp1487* and Δ*hp1489* mutants. (**A**) Motility of *H. pylori* B128 wild type (WT), *H. pylori* B128 Δ*hp1486*/*hp1487* mutant, and *H. pylori* B128 Δ*hp1489* mutant was assessed by inoculating *H. pylori* strains into soft agar medium and allowing the cells to migrate from the point of inoculation. (**B**) Motility of *H. pylori* G27 wild type (WT) and *H. pylori* G27 Δ*hp1489* mutant in soft agar medium. The two strains either had (+) or lacked (−) the shuttle vector pHel1489, which carried *H. pylori* G27 *hp1489* under control of its native promoter. Bars indicate the mean of swim halo diameters measured 7 d post-inoculation for at least 9 replicates. Error bars correspond to one standard deviation. For both panels (**A**,**B**), the asterisk (*) indicates a significant difference between the mutant and wild type (*p*-value < 0.0001). Statistical significance was determined using a two-sample *t*-test.

**Figure 3 ijms-23-11609-f003:**
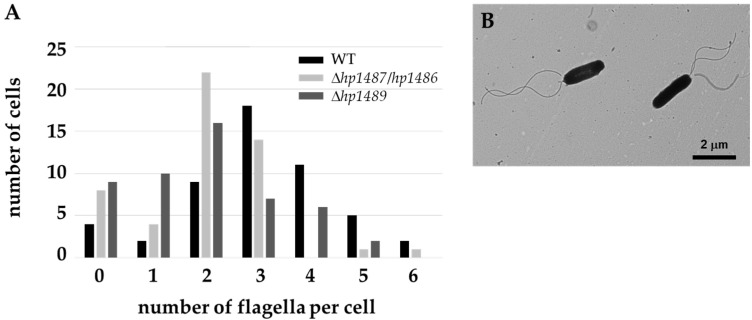
Flagella counts for *H. pylori* B128 wild type, Δ*hp1489* mutant, and Δ*hp1487*/*hp1486* mutant. (**A**) *H. pylori* cells were visualized by TEM and flagella were counted for 50 cells. Bar graph indicates number of cells that had the particular number of flagella per cell. Both mutants produced fewer flagella per cell than wild type (*p*-values < 0.01). Statistical significance was determined using a Mann-Whitney U test. (**B**) Typical *H. pylori* cells used for flagellar count data. Cell on the left has two flagella and cell on the right has three flagella.

**Figure 4 ijms-23-11609-f004:**
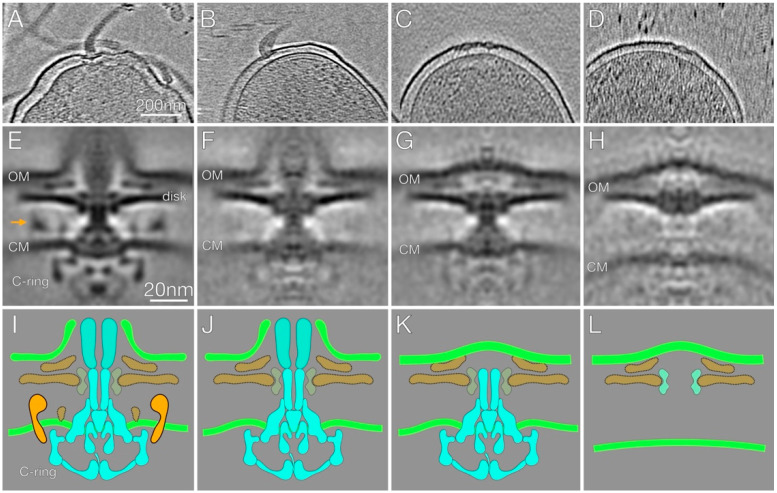
Visualization of flagellar motors of the *H. pylori* B128 Δ*hp1489* and Δ*hp1487*/*hp1486* mutants by cryo-ET. (**A**–**D**) Representative slices of cryo-ET reconstructions show various aberrant forms of flagellar motor from the mutants. (**E**–**H**) Central slices of the averaged structures of the specific classes of flagellar motors. Each averaged structure is below the representative slice for the specific class of motor. (**I**–**L**) The corresponding cartoon illustrations for the various classes of motor structures. The color scheme is: orange, cage; brown, PflA/PflB, P-disk and basal disk; gray, L ring and P ring; cyan, C ring, MS ring, export apparatus, rod, and hook; green, outer membrane/flagellar sheath; light green, inner membrane.

**Figure 5 ijms-23-11609-f005:**
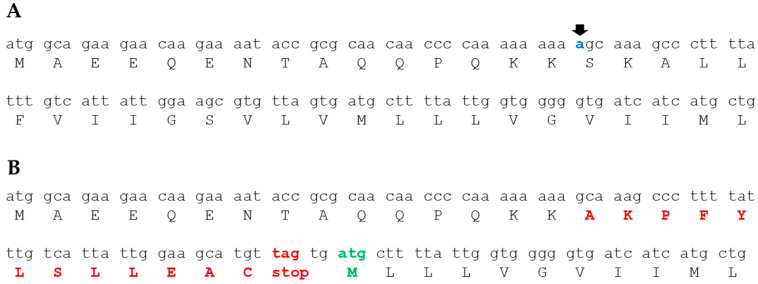
The *fliL* frameshift mutation in the motile variants of the Δ*hp1487*/*hp1486* mutant. (**A**) Nucleotide sequence of first 40 codons of wild-type *H. pylori* B128 *fliL* and translated amino acid sequence. The A residue that is deleted in all three Δ*hp1487*/*hp1486* motile variants is indicated in blue type and by the arrow. (**B**) Nucleotide and corresponding amino acid sequences of *fliL* from the Δ*hp1487*/*hp1486* motile variants. The altered amino acid sequence following the frameshift mutation is indicated in red type. A potential start codon that brings the nucleotide sequence into the correct reading frame is indicated in green type.

**Table 1 ijms-23-11609-t001:** Proteins conserved in *Helicobacter* species that have flagellar sheaths and underrepresented in *Helicobacter* species that have unsheathed flagella.

NCBI Reference	Locus Tag	Gene	Known/Proposed Function	Reference
**Flagellum function**				
WP_000742695.1	HP0327	*pseH*	flagellin glycosylation	[31]
WP_000868000.1	HPG27_395	*flgV*	required for motility	[32]
**Lipopolysaccharide biosynthesis**				
WP_000487418.1	HP0329	*futA*	fucosylation of LPS	[33]
WP_000433778.1	HP0580	*kdhA*	Kdo-lipid A hydrolase	[34]
WP_000898474.1	HP1580	*lpxF*	lipid A 4′-phosphatase	[35]
WP_041201341.1	HP1581	*wecA*	O-antigen assembly	[36]
**Sel1-like proteins**				
WP_000597817.1	HP0160	*hcpD*	penicillin-binding protein	[37]
WP_000901623.1	HP0211	*hcpA*	penicillin-binding protein	[38]
WP_000111740.1	HP0235	*hcpE*		
WP_000917816.1	HP0628	*hcpF*		
WP_000892789.1	HP1098	*hcpC*		
WP_000540103.1	HP1117			
WP_000943858.1	HPG27_1469	*hcpG*		
**Outer membrane proteins**				
WP_041201349.1	HP0209	*hofA*		[39]
WP_000768629.1	HP0486	*hofC*		[39]
WP_012552416.1	HP0487	*hofD*		[39]
WP_000911466.1	HP0782	*hofE*		[39]
WP_001108270.1	HP0788	*hofF*		[39]
WP_041201373.1	HP0914	*hofG*		[39]
WP_000797787.1	HP1083	*hofB*		[39]
**Transport proteins**				
WP_000788001.1	HP0839		fatty acid transport protein	[39]
WP_000816869.1	HP0970	*cznB*	metal efflux pump	[40]
WP_079990419.1	HP0971	*cznC*	metal efflux pump	[40]
WP_000780237.1	HP1028		lipocalin	[41]
WP_001008850.1	HP1486		ABC-2 family transporter protein	this study
WP_000489110.1	HP1487		ABC-2 family transporter protein	this study
WP_012552579.1	HP1488		membrane fusion protein	this study
WP_000754037.1	HP1489		outer membrane efflux protein	this study
**Miscellaneous functions**				
WP_000820014.1	HP0018		predicted lipoprotein	
WP_041201345.1	HP0097		predicted lipoprotein	
WP_000233964.1	HP0111	*hrcA*	heat-inducible repressor	[42]
WP_000689112.1	HP0190	*clsC*	cardiolipin synthase	[43]
WP_001159179.1	HP0199			
WP_000114771.1	HP0468		DUF5644 domain-containing protein	
WP_000462324.1	HP0640		poly(A) polymerase	
WP_000949206.1	HP0653	*ftnA*	bacterial non-heme ferritin	[44]
WP_000413451.1	HP0664		DUF2603 domain-containing protein	
WP_001279170.1	HP0700	*dgkA*	diacylglycerol kinase	
WP_000790557.1	HP0827		RNA or ssDNA-binding protein	
WP_001268507.1	HP0838		putative lipoprotein	
WP_012552558.1	HP1321		conserved hypothetical ATP-binding protein	
WP_000896338.1	HP1440	*ispDF*	isoprenoid biosynthesis	[45]

**Table 2 ijms-23-11609-t002:** Number of subtomograms for each motor class.

	Class 1	Class 2	Class 3	Class 4
Δ*hp1489* motor number	40	4	18	9
Δ*hp1487*/*hp1486* motor number	91	17	15	6

**Table 3 ijms-23-11609-t003:** Identified intragenic mutations in the Δ*hp1487*/*hp1486* motile variants.

Isolate	Gene Description	Locus Tag	Sequence ^a^	Impact ^b^	Freq ^c^
ESMV1	*arsS*	CV725_RS0308	(G)11→12 coding (1261/1281 nt)	P421fs	84.9%
	*fliL*	CV725_04935	(A)9→8 coding (46/552 nt)	S16fs	96.6%
	*oipA*	CV725_05780	(AG)9→8 coding (51-52/958 nt)	S18sf	93.3%
	glycosyltransferase family 25 protein	CV725_RS05870	(G)13→14 coding (539/633 nt)	T180fs	94.3%
	cag pathogenicity island protein	CV725_RS06350	2 bp→AG coding (2657-2658/5466 nt)	N886K	96.6%
ESMV2	cation:proton antiporter	CV725_RS04480	GCC→ACC	A249T	98.9%
	*fliL*	CV725_04935	(A)9→8 coding (46/552 nt)	S16fs	97.3%
	*murJ*	CV725_06605	(A)9→8 coding (1174/1462 nt)	S392fs	98.6%
	hypothetical protein	CV725_RS06835	CTA→CGA	L4R	100%
	hypothetical protein	CV725_RS06835	TTC→TCC	F5S	100%
	hypothetical protein	CV725_RS06835	2 bp→CT coding (17-18/2007 nt)	I6L	100%
	outer membrane beta-barrel protein	CV725_RS07750	(TC)9→8 pseudogene (14-15/1989 nt)	L5fs	96.4%
	glycosyltransferase family 8 protein	CV725_RS03290	(TC)9→8 pseudogene (95-96/1162 nt)	R28fs	96.2%
ESMV3	*fliL*	CV725_04935	(A)9→8 coding (46/552 nt)	S16fs	99.7%
	*murJ*	CV725_06605	(A)9→8 coding (1174/1462 nt)	S392fs	99.4%
	cation:proton antiporter	CV725_RS04480	GCT→ACT	A201T	99.1%

^a^ The numbers in parentheses indicates the position of the mutation (first number) within the entire length of the open reading frame (second number); ^b^ Indicates the amino acid position where a different amino acid was introduced or site where a frameshift mutation occurred (fs); ^c^ Indicates the percentage of reads at the position that had the particular mutation.

**Table 4 ijms-23-11609-t004:** *H. pylori* strains and plasmids used in this study.

**Strain**	**Relevant Genotype**	**Source**
B128	wild-type	[58]
G27	wild-type	[59]
HP116	*H. pylori* B128 *hp1489*:: kan^R^-P*_ureA_*-*sacB*	this study
HP119	*H. pylori* B128 ∆*hp1489*	this study
HP174	*H. pylori* B128 *hp1487/hp148639*:: *kan^R^-P_ureA_-sacB*	this study
HP185	*H. pylori* B128 ∆*hp1487*/*hp1486*	this study
HP121	*H. pylori* G27 *hp1489*:: kan^R^-P*_ureA_*-*sacB*	this study
HP122	*H. pylori* G27 ∆*hp1489*	this study
HP128	HP122 bearing pHel1489	this study
**Plasmid**	**Description**	**Source**
pGEM-T Easy	TA cloning vector; amp^R^	Promega
pJC038	pGEM-T Easy carrying kan^R^-P*_ureA_*-*sacB*	[43]
pJC049	pGEM-T Easy carrying flanking regions of *hp1489*	this study
pJC051	pJC049 with *kan^R^-P_ureA_-sacB* insertion	this study
pJC076	pGEM-T Easy carrying flanking regions of *hp1487*/*hp1486*	this study
pJC080	pJC076 with *kan^R^-P_ureA_-sacB* insertion	this study
pJC083	pGEM-T Easy carrying P*_hp1491_*-hp1489	this study
pHel3	*H. pylori* shuttle vector; kan^R^	[60]
pHel1489	pHel3 carrying *hp1489*	this study

## Data Availability

Not applicable.

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
