# Peer review of "A Tripartite Efflux System Affects Flagellum Stability in Helicobacter pylori"

_ijms, 2022, doi:10.3390/ijms231911609_

Round 1

Reviewer 1 Report

The authors of this paper have tried to elucidate the mechanism how the flagellar sheath is formed in H. pylori. Blastp picked four genes related with a tripartite efflux system as the candidates for components of the flagellar sheath formation. Deletion mutants of these genes gave rise to cells with fewer flagella and accordingly deterioration of swarm ability of the cells. Authors also isolated revertants from those deletion mutants and unexpectedly found that a modified FliL is responsible for the recovery of motility. Although it is still ambiguous to see how FliL is involved the flagellar sheath formation, the observation deserves publication.

I have several questions before accepting for print:

1. The authors claim that deletion mutants retain fewer flagella than the wild type (Fig.3). However, it is difficult for me to see the differences. TEM photos may help my understanding.

2. Have authors observed mutants swimming under the microscope? Changing the viscosity of media sometimes gives a difference of swimming between WT and mutants.

3. I know some of the authors are experts of cryo-EM and have published nice works on observation of flagellar motor by cryo-EM. However, I am not sure about existence of the PL-subcomplex in Fig.4.Even if thy do exist, how does the complex contribute to the sheath formation?

4.  What is the molecular size of FliL? The cleavage of the N-terminal region does not change the conformation of FliL at all?

5. Flagellar "disassembly" is a contradiction in terms.

Author Response

Comments from reviewer 1 and our responses to the comments (in red) are indicated below.

The authors of this paper have tried to elucidate the mechanism how the flagellar sheath is formed in H. pylori. Blastp picked four genes related with a tripartite efflux system as the candidates for components of the flagellar sheath formation. Deletion mutants of these genes gave rise to cells with fewer flagella and accordingly deterioration of swarm ability of the cells. Authors also isolated revertants from those deletion mutants and unexpectedly found that a modified FliL is responsible for the recovery of motility. Although it is still ambiguous to see how FliL is involved the flagellar sheath formation, the observation deserves publication.

I have several questions before accepting for print:

  1. The authors claim that deletion mutants retain fewer flagella than the wild type (Fig.3). However, it is difficult for me to see the differences. TEM photos may help my understanding.

An example of TEM image of H. pylori cells used for the flagellar count data has been provided in Figure 3B.

  1. Have authors observed mutants swimming under the microscope? Changing the viscosity of media sometimes gives a difference of swimming between WT and mutants.

We have not tried to observe and quantitate the swimming behavior of the H. pylori strains under a light microscope. Comparing the swimming behaviors of the H. pylori strains might provide some useful information, but we are not equipped to quantitate the swimming behavior of H. pylori.

  1. I know some of the authors are experts of cryo-EM and have published nice works on observation of flagellar motor by cryo-EM. However, I am not sure about existence of the PL-subcomplex in Fig.4.Even if they do exist, how does the complex contribute to the sheath formation?

Sheath biosynthesis was not inhibited in the Dhp1487-hp1486 or Dhp1489 mutants. We have clarified this point by including the sentence “Flagella of the Dhp1487-hp1486 and Dhp1489 mutants were sheathed, indicate sheath biosynthesis was not grossly impaired in the mutants.” (lines 159-161).

  1. What is the molecular size of FliL? The cleavage of the N-terminal region does not change the conformation of FliL at all?

We indicated the size of wild-type FliL and the size of the truncated FliL that would result from translation initiation at the potential internal start codon. (lines 274-276)

  1. Flagellar "disassembly" is a contradiction in terms.

This is not a term that we coined here. This was the term used by the authors of the papers cited in references 27 and 28.

Reviewer 2 Report

This is an admirable attempt to identify a potential ABC transporter, HP1486-HP1489, in H. pylori that affects its flagellum stability. Katherine Gibson and colleagues have identified this ABC transporter by comparing the proteome of flagellar sheaths and unsheathed flagella and targeted the tripartite efflux system using deletion analysis and cryo-ET. Their findings provide a better understanding of flagellum assembly and flagellar sheath biosynthesis in Helicobacter. Therefore, I recommend that this paper be published in the International Journal of Molecular Sciences. However, I have some reservations about the manuscript, detailed point by point below, that should be addressed:

It was found that four proteins are in one operon and are located at different locations on the membrane of H. pylori. There was no clear indication of whether these proteins formed a single tripartite system. It would be helpful if authors could provide evidence that those proteins form a transport system. This could be done by identifying the interaction through pull-down assays, colocalization them through microscopy, etc.

These four proteins do not contain an ATPase protein. There are some ABC prediction methods that may be able to predict an ABC transporter, such as ABC finder, DeepRTCP, or even alphafold prediction.

The statistical analysis of normal, incomplete motor, and filament-less mutant flagellums will provide an overview of flagellum assembly in those mutants. This analysis will clarify the role of those proteins in flagellum assembly.

In section 2.4, filL mutation could enhance mobility of the Δhp1487/hp1486 motile variants. To confirm filL's function, it would be better to induce mutations in filL alone since there are other intragenic mutations in the variants. In addition, inducing the filL mutation in Δhp1489 to study mobility will provide more clues about whether HP1487/HP1486 and HP1489 are part of the same system.

In Figure 3, it would be better to add a key to indicate what each bar refers to.

Minors:

In line 386, the phrase "Escherichia coli" should not appear in this line, but in the next line. The layout should be corrected before publication.

Author Response

This is an admirable attempt to identify a potential ABC transporter, HP1486-HP1489, in H. pylori that affects its flagellum stability. Katherine Gibson and colleagues have identified this ABC transporter by comparing the proteome of flagellar sheaths and unsheathed flagella and targeted the tripartite efflux system using deletion analysis and cryo-ET. Their findings provide a better understanding of flagellum assembly and flagellar sheath biosynthesis in Helicobacter. Therefore, I recommend that this paper be published in the International Journal of Molecular Sciences. However, I have some reservations about the manuscript, detailed point by point below, that should be addressed:

It was found that four proteins are in one operon and are located at different locations on the membrane of H. pylori. There was no clear indication of whether these proteins formed a single tripartite system. It would be helpful if authors could provide evidence that those proteins form a transport system. This could be done by identifying the interaction through pull-down assays, colocalization them through microscopy, etc.

The review raises a valid point about whether the four genes are involved in forma a single tripartite system. The synteny of the genes is conserved in the FS+ Helicobacter species, which strongly suggests the products of the genes participate in a common function. Nevertheless, we are currently working to tag HP1486 with a myc-tag for pull-down assays to identify the ATPase that functions with HP1486 and HP1487. We are also working to tag HP1486 with GFP for the localization studies the reviewer suggested.

These four proteins do not contain an ATPase protein. There are some ABC prediction methods that may be able to predict an ABC transporter, such as ABC finder, DeepRTCP, or even alphafold prediction.

We identified HP0853 as a good candidate for the ATPase that functions with the HP1487 and HP1486 ABC transporters. We included the following statement regarding HP0853 as a candidate for the ATPase in the Discussion. “The HP1489-HP1486 homologs in Helicobacter anseris MIT 04-9362 are encoded in an operon that contains a gene encoding a predicted ATPase for an ABC transport system. HP0853 is an orphan ATPase (i.e., the gene encoding the protein is not associated with genes encoding transmembrane components of an ABC transport system) and an ortholog of the H. anseris ATPase. We are currently investigating whether HP0853 is the ATPase that functions with HP1487 and HP1486.” (lines 332-337).

The statistical analysis of normal, incomplete motor, and filament-less mutant flagellums will provide an overview of flagellum assembly in those mutants. This analysis will clarify the role of those proteins in flagellum assembly.

We appreciate the suggestions from the reviewer. We did use multivariate statistical analysis to resolve distinct motor structures as shown in Figure 4. We now include additional Table (Table 2) to provide an overview of flagellar motor assembly in three mutants. We thank the reviewer for requesting the statistical information on the different classes of motor structures as we recognized that we had incorrectly reported that we did not observe class 2 and class 3 motors in the Dhp1487/hp1486 mutant. We have corrected this error in the manuscript (lines 225-226; 232-233; and 311-315).

In section 2.4, filL mutation could enhance mobility of the Δhp1487/hp1486 motile variants. To confirm filL's function, it would be better to induce mutations in filL alone since there are other intragenic mutations in the variants. In addition, inducing the filL mutation in Δhp1489 to study mobility will provide more clues about whether HP1487/HP1486 and HP1489 are part of the same system.

We assume the reviewer meant ‘introduce mutations in fliL’ rather than ‘induce mutations in fliL’. We are currently working on constructing these strains.

In Figure 3, it would be better to add a key to indicate what each bar refers to.

We included a key for the graph.

Minors:

In line 386, the phrase "Escherichia coli" should not appear in this line, but in the next line. The layout should be corrected before publication.

This has been corrected.

Round 2

Reviewer 2 Report

It is a remarkable work. Most of my reservations were addressed. To address all the questions, the work requires a great deal of effort. Your continued efforts will be greatly appreciated by the field.